# High-Resolution Network with Transformer Embedding Parallel Detection for Small Object Detection in Optical Remote Sensing Images

**Xiaowen Zhang** [1,2] **, Qiaoyuan Liu** [1,*] **, Hongliang Chang** [1,2] **and Haijiang Sun** [1]

1   Changchun Institute of Optics, Fine Mechanics and Physics, Chinese Academy of Sciences, Changchun 130033, China; zhangxiaowen22@mails.ucas.ac.cn (X.Z.); changhongliang22@mails.ucas.ac.cn (H.C.); sunhj@ciomp.ac.cn (H.S.)
2   University of Chinese Academy of Sciences, Beijing 100049, China
*   Correspondence: liuqy@ciomp.ac.cn

**Abstract:** Small object detection in remote sensing enables the identification and analysis of unapparent but important information, playing a crucial role in various ground monitoring tasks. Due to the small size, the available feature information contained in small objects is very limited, making them more easily buried by the complex background. As one of the research hotspots in remote sensing, although many breakthroughs have been made, there still exist two significant shortcomings for the existing approaches: first, the down-sampling operation commonly used for feature extraction can barely preserve weak features of objects in a tiny size; second, the convolutional neural network methods have limitations in modeling global context to address cluttered backgrounds. To tackle these issues, a high-resolution network with transformer embedding parallel detection (HRTP-Net) is proposed in this paper. A high-resolution feature fusion network (HR-FFN) is designed to solve the first problem by maintaining high spatial resolution features with enhanced semantic information. Furthermore, a Swin-transformer-based mixed attention module (STMA) is proposed to augment the object information in the transformer block by establishing a pixel-level correlation, thereby enabling global background–object modeling, which can address the second shortcoming. Finally, a parallel detection structure for remote sensing is constructed by integrating the attentional outputs of STMA with standard convolutional features. The proposed method effectively mitigates the impact of the intricate background on small objects. The comprehensive experiment results on three representative remote sensing datasets with small objects (MASATI, VEDAI and DOTA datasets) demonstrate that the proposed HRTP-Net achieves a promising and competitive performance.

**Keywords:** remote sensing; object detection; feature extraction; high-resolution; Swin transformer

## 1. Introduction

With the rapid development of remote sensing (RS), RS images with higher-resolution are able to show richer earth information, such as planes and ships under 20 m in length. Therefore, small object detection has become a hot topic in the RS field, which is of great significance to military object identification, traffic management, marine regulation, etc. [1–4]. Differently from detecting general objects with visually salient features in RS images, the task of small object detection is highly challenging: small objects often consist of only a few pixels, as shown in Figure 1a, while they may be easily obscured by intricate backgrounds, further increasing the difficulty of small object detection, as shown in Figure 1b–d. In Figure 1b, the small ship is surrounded by a complex background. The small objects in Figure 1c are disturbed by background illumination, and there is similar interference around the framed object in Figure 1d. The frameworks based on deep learning, such as faster R-CNN [5], SSD [6], and YOLO [7], have high efficiency in general object detection, but introducing these approaches directly into remote sensing tasks can hardly produce a

satisfactory performance. Therefore, detecting small objects in RS images requires more effort than the general object detection approaches.

Currently, most CNN-based detection approaches adopt a one-stream feature learning network that incorporates multiple down-sampling and up-sampling operations for context aggregation, such as U-Net [8] and FPN [9]. However, these networks discard details vital for small object detection. The proposed HRNET in [10] can compensate for the feature loss by maintaining the high-resolution feature representation in the forward propagation. HRNet is a representative method that is applied to tasks with complex information, such as general object detection [11], image segmentation [12], object tracking [13], etc. Differently from U-Net and FPN, HRNet utilizes parallel multi-resolution sub-networks for multi-scale repeated fusion. This approach enables HRNet to effectively fuse low-level information for precise localization and high-level semantic information to achieve strong robustness in occlusions and scale changes.

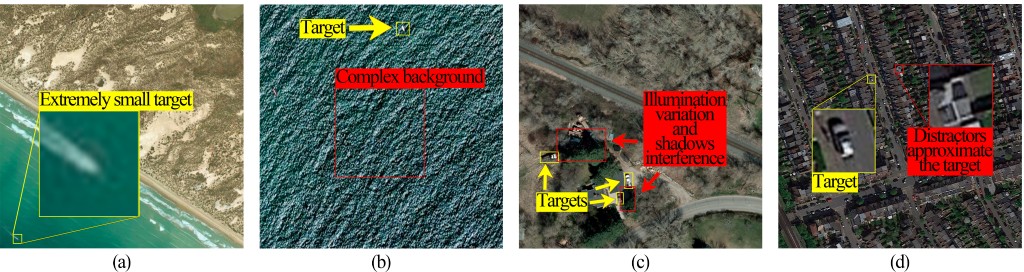

**Figure 1.** Examples of challenges in object detection in remote sensing images. (**a**) An extremely small object is in the bottom left corner of the image. (**b**) The ship object is surrounded by complex water waves. (**c**) Objects are disturbed by illumination variation and shadows. (**d**) There is similar interference around objects.

The methods based on CNN have shown excellent representation performance due to the locality and translation equivariance of the convolution operation. However, they struggle to model the global context information. In contrast, the self-attention approaches based on the transformer [14] have recently shown remarkable success for global modeling characteristics, which solve the locality limitation caused by the convolution. Ref. [15] explored the potential of the transformer in visual tasks and proposed a visual transformer (ViT) that completely discards the structure of CNN. Ref. [16] combined the transformer with YOLO for object detection in aerial images, which demonstrated the effectiveness of the transformer approach in small object detection. Liu et al. [17] proposed the Swin transformer and designed a multi-stage transformer that leverages a sliding window to reduce computational costs, which also proves the effectiveness of multi-scale features in the transformer. Although numerous studies have highlighted the effectiveness of the transformer in the remote sensing field [18–21], they lack the exploration between context information and local features.

Last but not least, many researchers are trying to combine the global modeling ability of the transformer with spatial or channel attention mechanisms to enhance the effectiveness of features [18,22]. However, when facing small targets, the global attention is more powerful when focusing on the limited receptive field, and these algorithms ignore the computational cost of the transformer. Therefore, we proposed a Swin-transformer-based mixed attention module (STMA) that leverages only the window-based transformer (W-trans) part [17], and a pixel-level attention module is attached to the W-trans to achieve background eliminating.

In this paper, to achieve the accurate detection of small objects in remote sensing images, a novel high-resolution network with transformer embedding parallel detection structure (HRTP-Net) is proposed. Differently from the existing small object detection methods, we abandon the traditional multi-sample operations. Instead, we proposed a high-resolution feature fusion network (HR-FFN) to maintain coordinates in space for

extremely small objects and tackle the drift location and preserve the weak features of small-scale objects during down-sampling. To alleviate the intricate obstruction elements in the complex background around the object, we further introduce STMA, which leverages a mixed attention module while generating a global connection to compensate for the correlation of multi-scale features generated by HR-FFN. In addition, we take the attentional features from the auxiliary detection structure with STMAs as a clue to guide the main detection structure, consisting of convolution layers. The main contributions of our work are given as follows:

1. A new multi-scale feature extraction network—HR-FFN—is proposed to retain more semantic information about small objects while achieving an accurate location, and we utilize the shallow and deep features simultaneously to alleviate the oversight of small-scale features during convolution operations.
2. To address the semantic ambiguity caused by background confusion, a mixed attention mechanism named STMA is proposed to enhance the distinguishable features by modeling global information to enhance the object saliency and establishing a pixel-level correlation to suppress the complex background.
3. A detection approach for small objects in remote sensing scenes, called HRTP-Net, is presented in this paper. Extensive ablation experiments show both the effectiveness and advantages of the HRTP-Net. The results on several representative datasets demonstrate that our approach brings significant improvement in small object detection.

## 2. Related Work

### 2.1. Multi-Resolution Feature Learning

Feature learning is one of the most necessary steps in general computer vision tasks, mainly based on two processes: (1) high-to-low process, which generates high-level but low-resolution representations; (2) low-to-high process, which recovers high-resolution representations from low-resolution representations. The feature learning networks could enhance the discriminability of objects by continuously repeating these two processes [10,23].

For the small object detection, multi-scale feature representation is the mainstream idea that is commonly used. The feature pyramid network (FPN) [9] was first proposed to learn multi-resolution features for multi-scale object detection. The path aggregation network (PANet) [24] extended FPN with a new down-top path that captures deeper features while exploiting the shallow features of the network. Additionally, the U-Net, which was originally designed for segmentation tasks, has also shown a great performance in detecting objects [25–27]. However, the down-sample operation often leads to the loss of semantic information when facing small objects in remote sensing images.

Hence, Sun et al. [10] proposed a high-resolution network (HRNet). Differently from recovering a high-resolution representation from a low-resolution representation, as discussed above, it can maintain a high-resolution representation during forward propagation. To implement remote sensing image segmentation, ref. [12] introduced HRNet as the backbone, and a feature-selection convolution (FSConv) layer was proposed for fusing the multi-resolution features, allowing adaptive feature selection based on object features. Ref. [28] added a csAG module consisting of spatial attention and channel attention to the HRNetv2 [11] network to minimize the error rate. The existing algorithms have taken advantage of the high-resolution representation of HRNet. Therefore, we generate the hypothesis that the high-resolution network will also be effective in small object detection.

### 2.2. Attention Mechanism in Deep Learning Network

The attention mechanism, drawing inspiration from human cognition, has already played an important role in computer vision [14,29]. Currently, the attention mechanism can mainly be divided into two branches: (1) use the pooling operation to extract salient information in the channel or spatial dimensions [29]; (2) use self-attention mechanisms to model global information and construct long-range dependencies [15].

The most representative approach in the first branch is the squeeze-and-excitation network (SENet) [30], which uses a global pooling and a fully connected layer to establish relationships between channels. Efficient channel attention (ECA) [29], based on the SENet, used 1D convolution instead of fully connected layers to avoid the compression of the channel dimension. In the convolutional block attention module (CBAM) [31], a spatial domain relevance calculation module was further proposed for selectively assigning importance to different features. Due to the lightweight and plug-and-play advantages, these methods are already widely used; however, their drawbacks in long-distance regions lead to the feature loss of small objects.

The transformer [14] is the most representative structure in the second branch and is able to extract features well in long-distance regions. To explore the potential of the transformer in computer vision, Alexey et al. [15] proposed a vision transformer (ViT) that turns images into tokens of sequences. However, the ViT still suffers from excessive computational effort and a low-resolution feature map. To solve these problems, the Swin transformer [17] is proposed to incorporate a window strategy to limit the computation cost of the self-attention system and adopt the window sliding mechanism of convolutional operations to realize the interaction between different windows, thus, achieving global attention. Although the Swin transformer has achieved remarkable results in a variety of tasks, this algorithm still has a large amount of computation; therefore, we simplify part of the Swin transformer structure. Zhang et al. [32] proposed ViT-YOLO, which combines YOLO with self-attention, leveraging its ability to process global object information. To address the challenge of fuzzy boundaries in remote sensing images, Xu et al. [18] added the dilated convolution module as a new backbone network based on the Swin transformer for a bigger receptive field. The above research explored the ability of the Swin transformer in modeling global information for object localization but disregarded the influence of a complicated background in remote sensing images. Therefore, we proposed a novel STMA structure to achieve a balance between performance and computational cost while alleviating the impact of the complex background.

### 2.3. Remote Sensing Images Object Detection

Currently, the CNN framework remains the mainstream framework for remote sensing object detection. The same as the generic object detection methods, object detection in remote sensing can also be divided into one-stage [6,7,33–37], and two-stage [5,38] algorithms. Specifically, the one-stage algorithms can achieve a faster detection with lightweight models. As the most representative approach in general detection tasks, YOLO is also the most widely used framework in remote sensing detection.

The small objects that only occupy a few pixels in satellite remote sensing images can easily be disturbed by complex backgrounds. Several studies have demonstrated that enhancing the multi-scale features can significantly improve small object detection. Etten et al. [39] proposed the You Only Look Twice (YOLT) algorithm, which applied YOLO to the remote sensing field for the first time. Han et al. [40] designed a multi-scale receptive field enhancement module (MRFEM) for small objects. Zhang et al. [41] proposed a multi-stage feature enhancement pyramid network to fuse features at varying scales for small objects with blurry edges. Popular attention mechanisms have also shown significant improvements in remote sensing detection. Kim et al. [42] proposed an efficient channel attention pyramid module based on YOLOv5 to improve the saliency of features for small objects. Hu et al. [43] proposed a spatial attention mechanism to optimize the feature representation of small objects and combined it with YOLOv5 for fast and accurate object detection. SuperYOLO [44] employed the super resolution learning and a symmetric compact multimodal fusion based on YOLOv5 to extract supplementary information from the various data. Taking advantage of the multi-scale feature fusion and attention mechanisms, Shi et al. [45] proposed FE-CenterNet, which uses a feature-enhanced module to extract multi-scale features and an attention-generation module to improve small object perception capability.

However, the existing studies have shown little attention to the resolution of feature maps and lack global modeling capabilities. In this approach, we aim to propose a new one-stage detection framework that better inherits the strengths of multi-scale features and attention mechanisms to handle small object detection under complex backgrounds in remote sensing images.

## 3. Proposed Method

### 3.1. Overall Structure

The framework of the proposed HRTP-Net is shown in Figure 2. We divide the overall model into four components, namely, the backbone, the high-resolution feature fusion network (HR-FFN), the auxiliary detection structure, and the main detection structure.

Given an input image $X \in \mathbb{R}^{H_1 \times W_1 \times 3}$, the spatial resolution of the image is reduced to 1/8 through the backbone to alleviate the computational pressure of the network. The output of the backbone is utilized to retain the semantic information in the HR-FFN module, which is described in detail in Section 3.2. In addition, during forward propagation, the high-resolution features interact with the lower-resolution features to convey object information. Then, the fused high-resolution (HR) features with abundant semantic information are passed to the auxiliary detection structure consisting of three STMA modules, which are described in detail in Section 3.4. The attention features obtained from the STMA are concatenated with HR features, and the C3 module is employed to aggregate features with different characteristics. Finally, the candidate boxes generated by the main detection structure are then subjected to non-max suppression (NMS) to obtain the detection results.

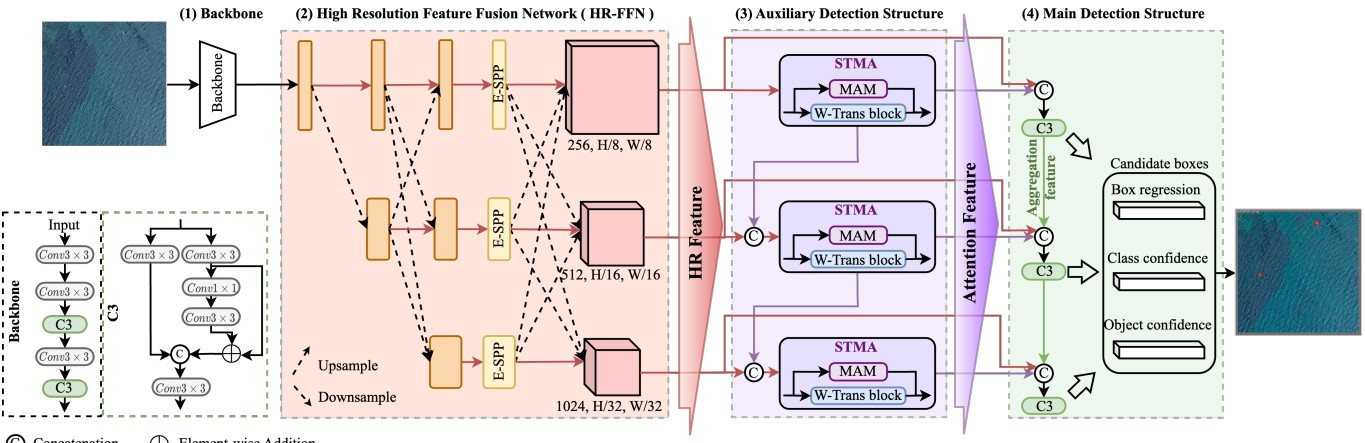

**Figure 2.** Overview of the proposed method HRTP-Net, which contains four main components, including: (**1**) the backbone for extracting basic high-resolution feature information; (**2**) HR-FFN for retaining the semantic information of high-resolution features, shown in Figure 3; (**3**) the auxiliary detection structure consist with the attention module STMA for enhancing feature effectiveness, shown in Figure 4; (**4**) the main detection structure for aggregating different features and final detection.

### 3.2. High-Resolution Feature Fusion Network (HR-FFN)

Considering the importance of semantic information in small object detection, the high-resolution feature fusion network (HR-FFN) is proposed, shown in Figure 3a, to better locate small objects. HR-FFN retains robust semantic information by employing parallel branches of multiple resolutions, maintaining the high resolution throughout the network. Additionally, the continuous information interaction between these branches further enhances its performance and compensates for the information loss caused by the decreasing channel dimensions. We divide the HR-FFN into three stages, with each stage adding one diverse resolution of the feature map over the previous stage as the forward propagation. The resolution of subnetwork $n$ can be expressed as $H_n \in \mathbb{R}^{(H/2^{n+2}) \times (W/2^{n+2}) \times (2^{n-1}C)}$ where

$n = 1, 2, 3$, and $C = 256$. The network generates high-resolution images by nearest neighbor sampling and obtains low-resolution images by a convolution operation with a $3 \times 3$ kernel.

In the HR-FFN, the input of the basic block is $B_{in} \in \mathbb{R}^{H \times W \times C}$, and the output $B_{out} \in \mathbb{R}^{H \times W \times C}$ can be expressed by the following equation:

$$B_{out} = B_{in} + f(f(B_{in})), \tag{1}$$

where $f(\cdot)$ denotes the convolutional layer with batch normalization and the ReLU activation function. With the fuse layer, the multi-resolution features first go through a concatenation operation to obtain a feature map $F$. The feature map $F$ is handled with convolution and batch normalization operations to generate the output, where the output channel varies with the position of the fuse layer.

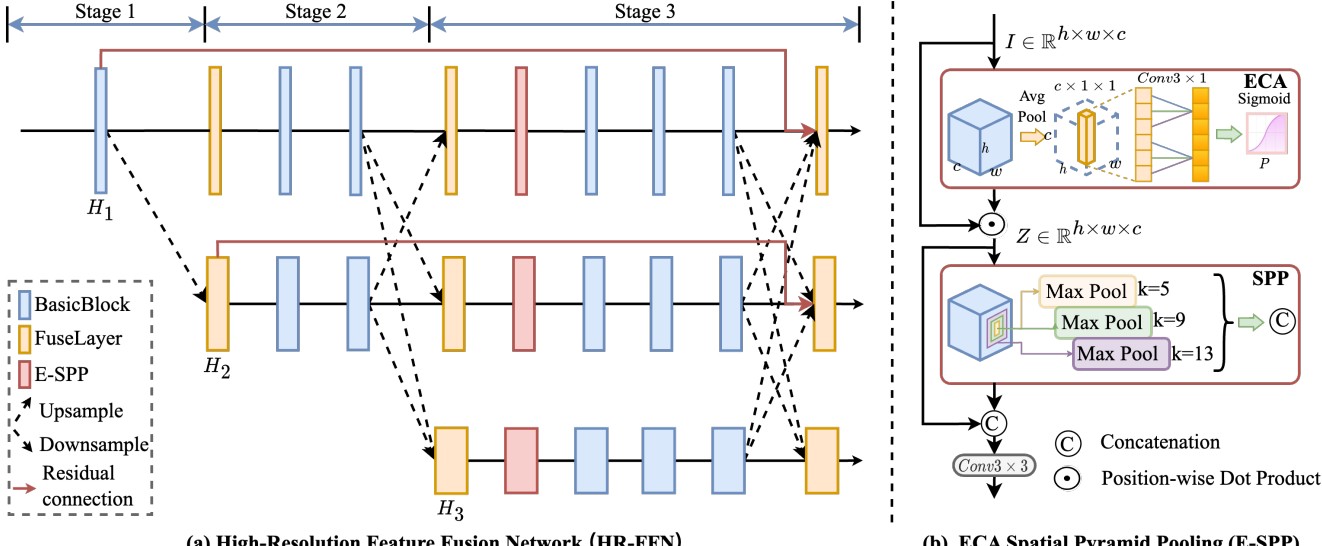

**(a) High-Resolution Feature Fusion Network (HR-FFN)**          **(b) ECA Spatial Pyramid Pooling (E-SPP)**

**Figure 3.** (**a**) Proposed structure of HR-FFN. HR-FFN consists of a basic block, fuse layer, and E-SPP. The black arrow represents the network forward propagation. The red arrow in the figure indicates the residual connection that combines the shallow and deep features. (**b**) Structure of ECA Spatial Pyramid Pooling (E-SPP), which consists of an ECA module and an SPP module.

We found that the down-sampling operation in the HRNet still results in feature loss for small objects, so we added the ECA spatial pyramid pooling (E-SPP) module in the third stage to mitigate feature weakening. We reweight the multi-channel information contained in feature maps by ECA. Then, the SPP module facilitates the retrieval of spatial features with different scales, thereby enhancing the robustness of the model to spatial layout and object variability. The E-SPP module can focus on the relevant information and prevent the loss of the fine-grained details. As shown in Figure 3b, the E-SPP module consists of two parts, the efficient channel attention (ECA) [29] mechanism and the spatial pyramid pooling (SPP) [46] module. In ECA, the input feature map is $I \in \mathbb{R}^{H \times W \times C}$, and a global average pooling and a 1D convolution of kernel size $k = 3$ are performed to generate channel weights. Then, an activation function sigmoid is used to obtain a feature map $P \in \mathbb{R}^{1 \times 1 \times C}$. Finally, the input feature map $I$ and the feature map $P$ are multiplied by $\odot$ to obtain the channel attentional output $Z \in \mathbb{R}^{H \times W \times C}$, where $\odot$ stands for position-wise dot production. The ECA is lightweight and efficient; therefore, we also use the ECA in the STMA module, which is detailed in Section 3.4.

The output of the ECA is used as the input for the SPP. In the SPP, the input $Z$ goes through max pooling with three different kernel sizes (5, 9, and 13), respectively. Afterwards, these feature maps are concatenated with the input $Z$. In small object detection, we found that shallow features in the network are equally important compared to maintaining high-resolution forward propagation. Therefore, we fused the shallow features in the

network with the deep features of stage 3 to obtain feature maps with rich information, as shown in red dashed arrows.

### 3.3. Window-Based Transformer (W-Trans)

The transformer is able to obtain global information and long-distance connections in one single operation. In the self-attention process, the input matrix is first expanded in the channel dimension and sliced by a linear layer to obtain three matrices, query ($Q$), key ($K$), and value ($V$); each of them can be expressed as $Q, K, V \in \mathbb{R}^{H \times W \times C}$. The output of the self-attention module is given by the following equation:

$$Attention(Q, K, V) = Softmax(\frac{QK^T}{\sqrt{(d_k)}})V, \tag{2}$$

where $d_k$ is the number of channels of matrix $K$. Dividing the inner product value with $\sqrt{(d_k)}$ is to prevent the vectors from being too large.

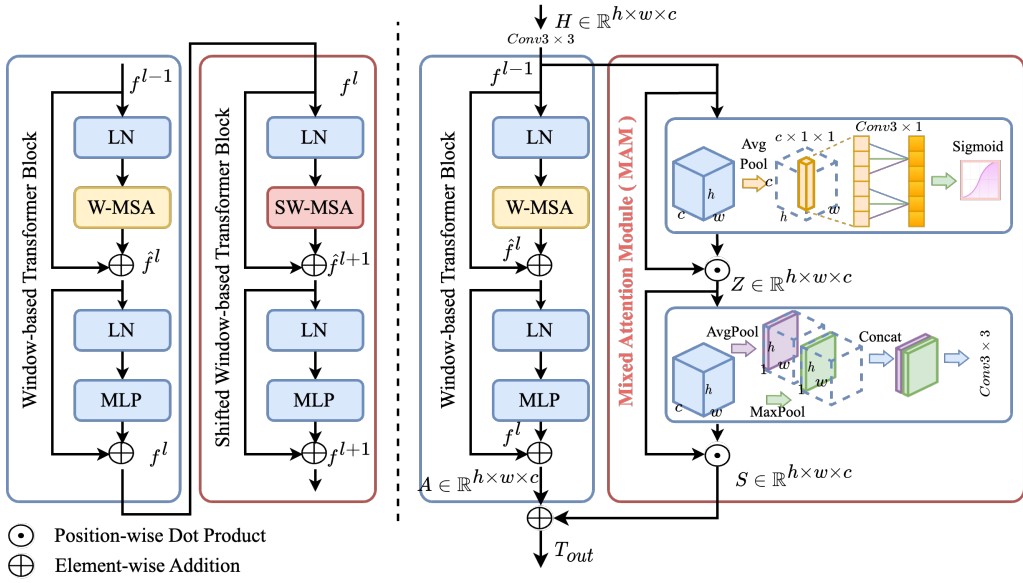

**(a) Standard Swin transformer block**　　　**(b) Swin transformer based mixed attention module (STMA)**

**Figure 4.** (**a**) Standard Swin transformer block [17]. (**b**) Structure of STMA, consisting of window-based transformer block and mixed attention module (MAM). Differently from the standard Swin transformer block, we replace the shifted window-based transformer block with our proposed MAM to enhance the interaction between windows.

Recently, the Swin transformer achieved an advanced performance in various vision tasks such as image classification, object detection, and semantic segmentation [22,47,48]. The Swin transformer proposes an efficient modeling solution with two window configurations, namely, the window-based multi head self-attention (W-MSA) and shifted window-based multi head self-attention (SW-MSA) models, as shown in Figure 4a. First, the Swin transformer divides the input feature map into windows in the window-based transformer block (W-trans). Then, in the shifted window-based transformer block (SW-trans), the windows are translated to enable the information interaction between the windows in W-trans. Each window is subjected to the operation of the self-attention mechanism, where each window covers only $D \times D$ pixels of the image. In our experiments, we set $D$ to 8 for the computational convenience. W-trans and SW-trans are executed alternately in successive Swin transformer blocks to enhance the information connectivity across windows. We express the basic blocks of the Swin transformer in the following equation:

$$\hat{f}^l = W\text{-}MSA(LN(f^{l-1})) + f^{l-1},$$
$$f^l = MLP(LN(\hat{f}^l)) + \hat{f}^l,$$
$$\hat{f}^{l+1} = SW\text{-}MSA(LN(f^l)) + f^l, \tag{3}$$
$$f^{l+1} = MLP(LN(\hat{f}^{l+1})) + \hat{f}^{l+1}.$$

The Swin transformer employs a uniform-size window to establish connections among its internal pixels, thereby effectively reducing memory consumption compared to the ViT. However, the W-trans and SW-trans still require a huge amount of computational resources compared to regular convolutional operations. Additionally, in small object detection in remote sensing images, there is often a significant size difference between the object and the entire image. Therefore, we leverage the window mechanism in the Swin transformer to localized regions for self-attention instead of processing the entire image. We simplify the original Swin transformer by retaining only the W-trans structure as part of the STMA, as shown in Figure 4b. Additionally, we implemented a mixed attention module (MAM) to reduce the influence of intricate background information. The W-trans and MAM enable the proposed STMA to effectively model the background–object information, which is described in the next section.

### 3.4. Swin-Transformer-Based Mixed Attention Module (STMA)

Remote sensing images often contain complex background information that can interfere with object detection, which requires some spatial information to alleviate the impact. Therefore, we propose STMA, consisting of a W-trans and mixed attention module (MAM), as shown in Figure 4b. The W-trans is responsible for encoding the global information features and accurately localizing small objects. The mixed attention module (MAM) is able to suppress background interference by making full use of the attention mechanism to create a pixel-level correlation and highlight important and representative channels from the whole feature map. In STMA, the input feature map is initially processed by the W-trans and MAM modules to model the object and background information separately. Afterwards, the outputs of the W-trans and MAM modules are combined through element-wise addition to obtain the background–object modeling. Through this process, the STMA helps to suppress complex information from the background, enhancing the discriminative power for object detection.

The CNN-based feature extraction modules extract local information in the spatial dimension limited by convolutional kernels but lack modeling of the relationships in the channel dimension [30]. Several approaches (e.g., [29,31]) have demonstrated that encoding the dependencies in the channel dimension can improve the feature extraction. Therefore, we designed a mixed attention module via two other attention modules, channel and spatial. The experiments in this research [31] demonstrated that a sequential arrangement of attention modules gives better results than a parallel arrangement. In STMA, the input expressed as $H \in \mathbb{R}^{h \times w \times c}$ goes through a convolutional operation and is split into two branches, one through the MAM module and the other through the transformer block.

In the previous section, we described the structure of ECA, and in this section, ECA is also used for information aggregation in the channel dimension. The output of the feature map after ECA processing is $Z \in \mathbb{R}^{h \times w \times c}$. Then, the output $Z$ is applied to obtain the spatial statistics of the feature map. In computing the spatial attention, we first apply the average pooling and max pooling operations with a $3 \times 3$ kernel along the channel dimension—the resulting feature maps are $E_A \in \mathbb{R}^{h \times w \times 1}$ and $E_M \in \mathbb{R}^{h \times w \times 1}$—and connect them to generate a valid descriptive feature. Applying the pooling operation along the channel axis can effectively highlight the information region, and the output of the pixel-level MAM is $S \in \mathbb{R}^{h \times w \times c}$. This process can be represented by the following equation:

$$S = \varphi(cat(E_A, E_M)) \odot Z, \tag{4}$$

where cat denotes the concatenate operation of the feature map along the channel after the pooling operation. $\varphi(\cdot)$ denotes a convolution operation and a ReLU activation function. $\odot$ stands for matrix multiplication. For a given input feature map $H$, the output is $A \in \mathbb{R}^{h \times w \times c}$ via the window-based transformer block. Finally, the global feature of self-attention is combined with the local feature of the MAM to form the output of STMA, which can be expressed as:

$$T_{out} = S \oplus A, \tag{5}$$

where $\oplus$ stands for element-wise addition. Eventually, the global cues with the channel information extracted from the auxiliary detection structure composed of STMA modules are used to guide the main detection structure composed of convolutional layers, achieving a complementary performance by the attention mechanism and the convolutional layers.

## 4. Experiments

In this section, we test the proposed method on the MASATI [49], VEDAI [50], and DOTA [51] datasets. Precision (P), recall (R), average precision (AP), and frames per second (FPS) are selected as evaluation metrics. In addition, we performed ablation experiments to verify the effectiveness of each module. Finally, the experimental results demonstrate the superiority of the proposed method.

### 4.1. Datasets

#### 4.1.1. MASATI Dataset

The MASATI dataset [49] contains a total of 7389 images and 3951 objects, including different marine scenes such as pure sea, reef, and port. These images include single or multiple ships under varying background conditions with a size of $512 \times 512$ pixels. In the experiments, 2368 images containing objects are selected and classified into three categories according to their characteristics, namely, multi-ship, single-ship, and coast.

#### 4.1.2. VEDAI Dataset

The VEDAI dataset [50] includes small objects from remote sensing images in a wide variety of environments. In the VEDAI dataset, the objects are small and present varying degrees of variability. In addition, different types of backgrounds can be found in this dataset, including urban, peri-urban, rural areas, etc. The images with a resolution of $512 \times 512$ pixels have a spatial resolution of 25 cm, while the object size ranges between 8 to 20 pixels. The dataset contains 3757 objects belonging to 9 different categories: plane, boat, camping car, car, pick up, tractor, truck, van, and others.

#### 4.1.3. DOTA Dataset

The images in the DOTA dataset [51] are collected from Google Earth, GF-2, and the JL-1 satellite provided by the China Center for Resources Satellite 585 Data and Application. The image size varies from $800 \times 800$ pixels to $20,000 \times 20,000$ pixels, and DOTAv1.5 is used in the experiments, which contains a total of 403,318 instances, including 16 categories such as plane, ship, storage tank, baseball diamond, tennis court, basketball court, ground track field, harbor, bridge, large vehicle, small vehicle, helicopter, roundabout, soccer ball field, swimming pool, and container crane. In this paper, two of these categories (i.e., small vehicle and ship) are chosen to train the models, and the images are horizontally cropped by 1000 pixels with a 10% overlap.

Following [52], we use the "object image ratio" to describe the relative size of an object in the image, which is $Ratio = O_s / I_s$, where $O_s$ is the area of the object, $I_s$ is the area of the image, $O_s = O_w \times O_h$ pixels, and $I_s = I_w \times I_h$ pixels. We refer to $Ratio \in (0, 0.001)$ as a small object and $Ratio \in (0, 0.01)$ as a medium object. As shown in Figure 5, we count the object-to-image ratios in different datasets. In the MASATI and VEDAI datasets, the percentage of small objects and medium objects is approximate, with slightly more medium objects in the VEDAI dataset. In DOTA, on the other hand, more than 80% of the objects are small objects, which makes it more difficult for object detection.

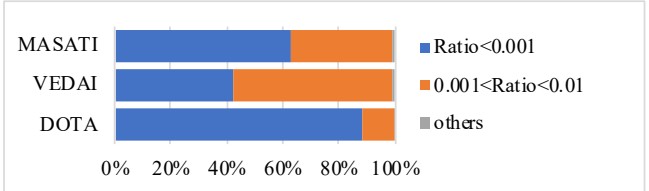

**Figure 5.** The proportion of images with different "object image ratio" in the MASATI, VEDAI, and DOTA datasets.

## 4.2. Implementation Details

### 4.2.1. Evaluation Metrics

We quantitatively evaluate the performance of the proposed method through several key evaluation metrics, including precision (P), recall (R), mean average precision (mAP), and frames per second (FPS). These evaluation indexes are considered to be the most widely used metrics in object detection applications. The P and R are defined as:

$$
\begin{aligned}
Precision &= \frac{TP}{TP + FP}, \\
Recall &= \frac{TP}{TP + FN},
\end{aligned}
\tag{6}
$$

where TP, FP, and FN stand for the number of accurately detected objects (true positive), false alarms (false positive), and missing objects (false negative). Additionally, mAP denotes the average precision value over different recalls, which is obtained by calculating the area under the precision–recall curve based on the corresponding precision and recall values.

$$
mAP = \int_0^1 P(R)\, \mathrm{d}R,
\tag{7}
$$

where $P(R)$ is a function of P and R. Furthermore, we apply the frames per second (FPS) indicator to evaluate the real-time detection efficiency.

### 4.2.2. Training Settings

In this paper, all experiments and models are built using the PyTorch framework with Intel i9-13900k, GeForce RTX4090 GPU, and Ubuntu 22.04. In the experiments, the optimizer is stochastic gradient descent (SGD), which utilizes an initial learning rate of $1 \times 10^{-2}$, a weight decay parameter of $5 \times 10^{-4}$, and a momentum of 0.937. In the initial stage of model training, 3 epochs are used for the warm-up training. Moreover, to enhance the data diversity and enrich the background information of the images, a mosaic data enhancement method is utilized to combine four images into one, thus, increasing the batch size. We selected 60% of the data from MASATI for training, 20% for validation, and the final 20% for the test. We sampled the same training and testing process in the DOTA and VEDAI datasets as well. In addition, we set the threshold for intersection over union (IoU) to 0.6 for a fair comparison of the methods.

## 4.3. Ablation Experiments

In this section, to measure the contribution of each proposed structure to the results, we conducted ablation experiments on the MASATI dataset. Compared with the DOTA and VEDAI datasets, the MASATI dataset includes single-object and multi-objects with complex scenarios, which can better evaluate the effect of the proposed modules. The baseline used in the experiments is the YOLOv5 algorithm. The contribution to the final results is analyzed by assessing the detection performance before and after the use of the module. The results of our ablation experiments are presented in Table 1. "✓" indicates that the module is used in the HRTP-Net, while "-" indicates that the module is not used in the HRTP-Net.

**Table 1.** Ablation experiment of the proposed modules on the MASATI dataset (The best performance is highlighted with bold font).

| Model | Structure | | | Multi-Ship | Single-Ship | Coast | All Classes | | |
| | HR-FFN | STMA | Auxiliary Detection Structure | mAP (%) | | | Precision (%) | Recall (%) | mAP (%) |
|---|---|---|---|---|---|---|---|---|---|
| Baseline1 [1] | - | - | - | 80.7 | 72.9 | 62.4 | 78.1 | 70.0 | 70.9 |
| HRTP-Net | ✓ | - | - | 81.3 | 81.4 | 65.0 | 82.5 | 72.5 | 75.0 |
| | - | ✓ | - | 78.8 | 80.2 | 64.7 | 82.2 | 67.4 | 73.4 |
| | - | ✓ | ✓ | 80.9 | 80.6 | **68.7** | 80.3 | 71.1 | 75.6 |
| | ✓ | ✓ | - | 79.8 | **85.7** | 65.3 | 82.0 | **74.0** | 75.7 |
| | ✓ | ✓ | ✓ | **84.3** | 82.8 | 67.0 | **84.6** | 72.0 | **77.3** |

[1] The baseline1 in Table 1 denotes the YOLOv5 algorithm.

### 4.3.1. Effect of High-Resolution Feature Fusion Network

The HR-FFN is proposed to address the problem of small objects with loss of detail during feature extraction. The second row in Table 1 shows the effect of HR-FFN on the detection performance when added to the baseline. The improvement in mAP is 0.6% for multi-ship, 8.5% for single-ship, and 2.6% for coast. Eventually, there is an effective improvement of 4.5%, 1.9%, and 4.1% on precision, recall, and mAP in the detection performance of these categories compared to the baseline for all categories. The detection of these small objects can be influenced by network down-sampling. Figure 6 shows comparison of the detection results before and after using HR-FFN in the HRTP-Net.

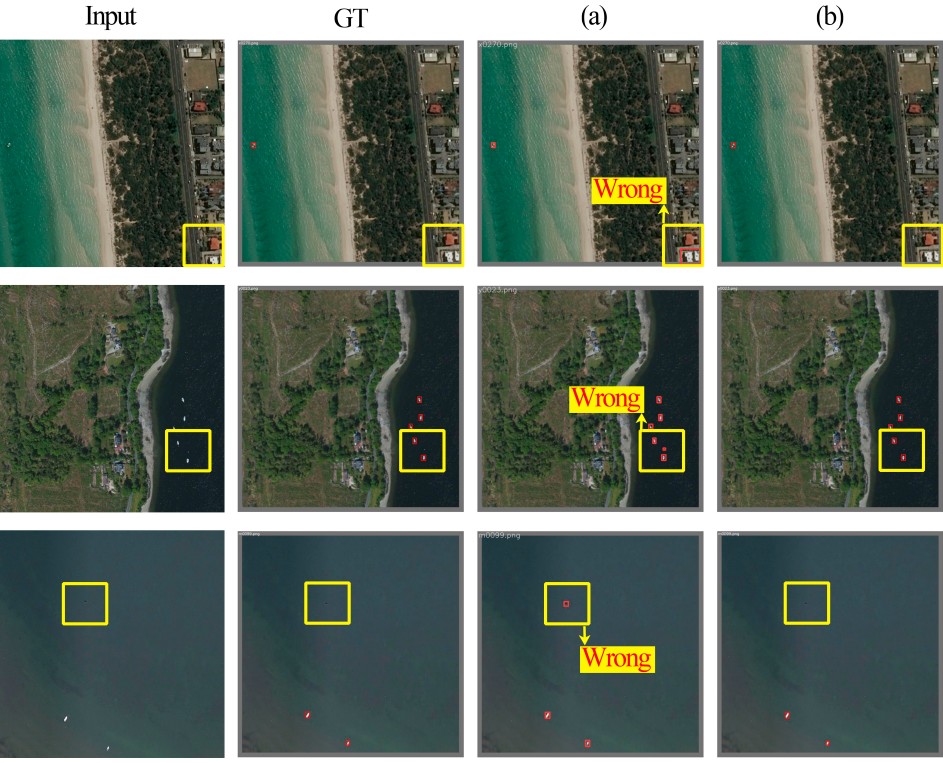

**Figure 6.** Comparison of the detection results before and after using HR-FFN in the HRTP-Net. (**a**) Baseline. (**b**) Baseline + HR-FFN.

We also designed an ablation experiment on the HR-FFN to test the effects of our proposed E-SPP and the residual connection on the model, respectively. From the second row of Table 2, we can see that the model rises by 2.4% for precision, 0.7% for recall, and

1.9% for mAP when E-SPP is added compared to the baseline. The E-SPP module is able to reassign weights to channel dimensions and extract information from feature maps with different receptive fields, thus, achieving a significant improvement in the precision index. After adding the residual connections, as in the third row of Table 2, there is a 0.8% improvement on recall and a 0.3% increase in mAP but a decrease in the precision metric. We suspect that the residual connections bring in small object features as well as interference information. Finally, the combination of the E-SPP module and the residual connectivity effectively improved the HR-FFN by 3.6%, 1.8%, and 1.7% for precision, recall, and mAP.

**Table 2.** Ablation experiment for the HR-FFN (The best performance is highlighted with bold font).

| Model | Structure | | All Classes | | |
|---|---|---|---|---|---|
| | E-SPP | Residual Connection | Precision (%) | Recall (%) | mAP (%) |
| Baseline2 [1] | - | - | 78.9 | 70.7 | 73.3 |
| HRTP-Net | ✓ | - | 81.3 | 71.4 | **75.2** |
| | - | ✓ | 76.7 | 71.5 | 73.6 |
| | ✓ | ✓ | **82.5** | **72.5** | 75.0 |

[1] The baseline2 in Table 2 denotes the proposed method HRTP-Net without E-SPP and residual connection.

### 4.3.2. Effect of Swin-Transformer-Based Mixed Attention Module

The STMA proposed in HRTP-Net aims to enhance the global information in the transformer block by establishing a pixel-level correlation to strengthen local feature effectiveness. It is important to mention here that when the model does not use an auxiliary detection structure but uses STMA, we replace the C3 module in the main detection structure with STMA for the experiment. Based on this explanation, there is only the main detection structure consisting of STMA existing in the model in the third and fifth rows of Table 1. Ultimately, when the STMA is used in the network, the detection results show a 4.1% improvement in precision and a 2.5% improvement in mAP. We also visualize the effect of STMA, as shown in Figure 7; for single-object, multi-object, and complex background cases, the model with STMA achieves a more effective feature extraction than without the inclusion of STMA.

We also analyze the effectiveness of the MAM module. We use models with the addition of HR-FFN and the auxiliary detection structure for comparison; the difference is whether or not we add MAM to the STMA module. In the first row of Table 3, the model using the basic window-based transformer block structure achieves 82.9%, 70.9%, and 75.4% in precision, recall, and mAP. The model in the second row of the table uses the full STMA module. It can be seen that with the addition of the MAM module, the model outperforms the W-trans structure by a large amount in the precision with 1.7%, in the recall with 1.1%, and the mAP with 1.9%.

**Table 3.** Ablation experiment for the STMA (The best performance is highlighted with bold font).

| Model | Structure | All Classes | | |
|---|---|---|---|---|
| | MAM | Precision (%) | Recall (%) | mAP (%) |
| HRTP-Net | - | 82.9 | 70.9 | 75.4 |
| | ✓ | **84.6** | **72.0** | **77.3** |

### 4.3.3. Effect of Auxiliary Detection Structure

In our design, the auxiliary detection structure serves the STMA module, so when the auxiliary detection structure is employed, the STMA module is also employed. As shown in the third and fourth rows of Table 1, the model has a 2.2% improvement in mAP after adopting the auxiliary detection structure. Similarly, comparing the fifth row with

the sixth row, the model has a significant improvement in mAP by 1.6% after adopting the auxiliary detection structure. We find that the adoption of the auxiliary detection structure effectively improves the model's mAP but has the opposite effect on the precision and recall metrics. From the experimental results in Table 1, we conclude that the auxiliary detection structure using STMA is able to compensate for the limitations of the convolutional layer in the main detection structure through the attention mechanism, resulting in a significant improvement in the overall detection effectiveness of the network by the mAP index.

The joint effect between the modules in the HRTP-Net framework is shown in Table 1. With the simultaneous introduction of STMA and the auxiliary detection structure, precision, recall, and mAP increased by 2.2%, 1.1%, and 4.7%, respectively. When HR-FFN and STMA are considered simultaneously, the detection results improved by 3.9% for precision, 4.0% for recall, and 4.8% for mAP. Our HRTP-Net with three components (HR-FFN, STMA, the auxiliary detection structure) brings an increase of 6.5% in precision, 2.0% in recall, and 6.4% in mAP compared with the baseline.

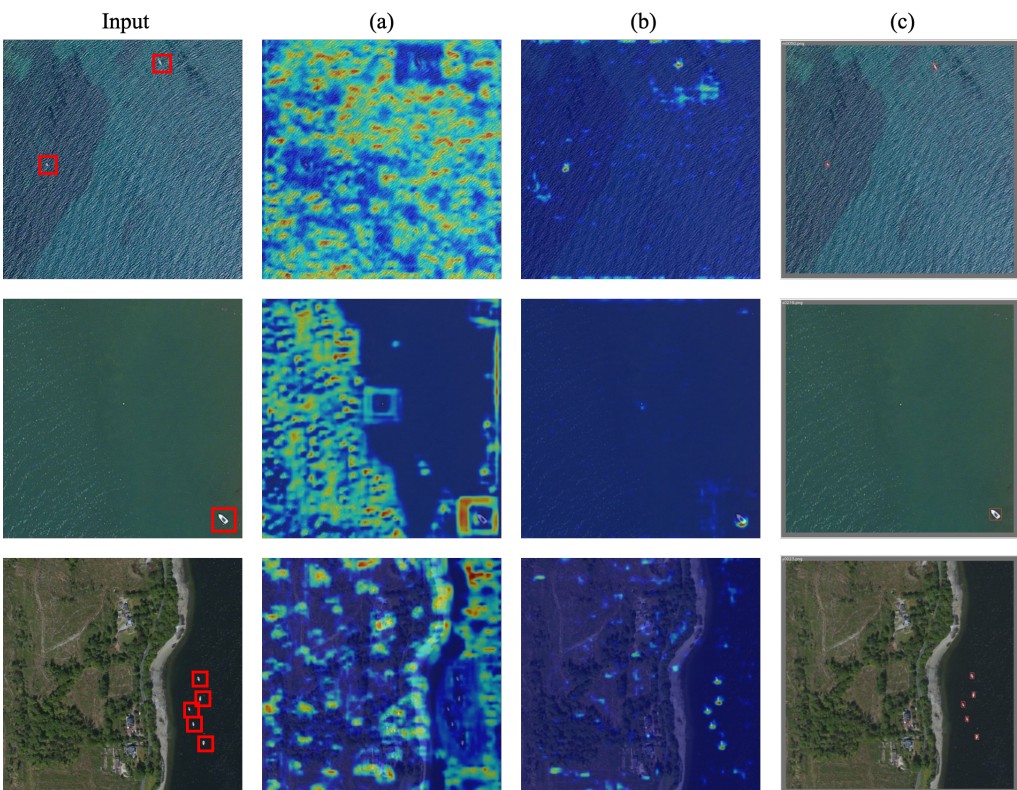

**Figure 7.** Visualization of feature maps and detection results. (**a**) Visualization of feature maps produced by the model without STMA. (**b**) Visualization of feature maps produced by the model with STMA. It can be seen that STMA can eliminate most obstructed backgrounds. (**c**) Detection results with baseline+auxiliary detection structure.

*4.4. Comparison With Other Methods*

We compare the proposed HRTP-Net with a number of existing methods, including YOLOR [33], YOLOv4 [34],YOLOv5 [37], YOLOv7 [35], PicoDet [36], SuperYOLO [44], PAG-YOLO [43], THP-YOLOv5 [16], Faster R-CNN [5], Cascade R-CNN [38], and HRNet [11]. The first eight methods are one-stage approaches and the last three methods are two-stage approaches. Our proposed HRTP-Net is a one-stage method. PAG-YOLO [43] and TPH-YOLOv5 [16] are algorithms using spatial and self-attention mechanisms, respectively. In addition, we do not use the extra prediction head to run the TPH-YOLOv5. Both the Faster R-CNN and Cascade R-CNN use RetNet50 as the backbone, and the two-stage models use a pre-trained backbone network in order to reduce the training time.

4.4.1. Results on MASATI Dataset

We compare our method with some of the most advanced detectors, as shown in Table 4. The improved HRTP-Net achieves an mAP of 77.3% in the MASATI dataset, which exceeds many of the current state-of-the-art methods. Figure 8 shows the objects detected by our method with different backgrounds on the MASATI dataset. As we expected, HRNet achieved the highest mAP in the single ship category with 83.9%, but its performance is comparatively lower in classes such as multi-ship and coast, which entail multiple objects and land disturbances. This validates our hypothesis in the previous section that HRNet is able to retain more detailed features, which is better for small object detection but also introduces more background information to the interference. In the results for all categories, our proposed algorithm in this paper far outperforms the other algorithms in precision and mAP, but the recall metric is slightly worse than HRNet, which is a problem we will have to address in the future.

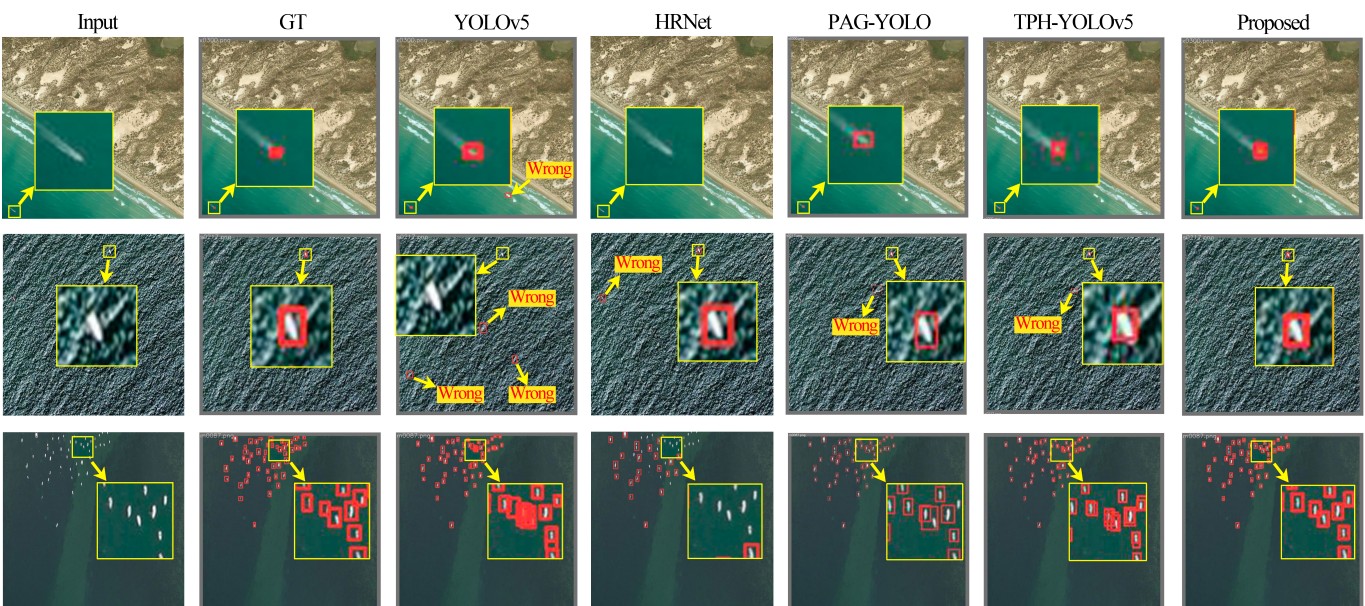

**Figure 8.** Examples of detection results on the MASATI dataset using different methods involving YOLOv5, HRNet, PAG-YOLO, TPH-YOLOv5, and the proposed method.

**Table 4.** Comparison of detection results from the MASATI dataset (The best performance is highlighted with bold font).

| Model | Multi-Ship | Single-Ship | Coast | All Classes | | |
|---|---|---|---|---|---|---|
| | mAP (%) | | | Precision (%) | Recall (%) | mAP (%) |
| YOLOR [33] | 75.7 | 64.1 | 52.7 | 75.9 | 66.4 | 63.1 |
| YOLOv4 [34] | 79.4 | 74.0 | 61.5 | 76.8 | 69.0 | 70.5 |
| YOLOv5 [37] | 80.7 | 72.9 | 62.4 | 78.1 | 70.0 | 70.9 |
| YOLOv7 [35] | 77.7 | 81.2 | 59.3 | 79.6 | 70.5 | 71.3 |
| PicoDet [36] | 81.7 | 61.1 | 68.4 | 80.1 | 64.3 | 70.1 |
| Faster R-CNN [5] | 70.0 | 77.2 | 38.7 | 77.6 | 64.4 | 59.2 |
| Cascade R-CNN [38] | 77.4 | 76.8 | 58.5 | 77.5 | 62.1 | 67.2 |
| HRNet [11] | 69.4 | **83.9** | 49.1 | 70.4 | **74.1** | 66.3 |
| SuperYOLO [44] | 80.3 | 73.4 | 65.5 | 76.7 | 70.5 | 72.7 |
| PAG-YOLO [43] | 82.8 | 81.9 | 65.8 | 83.3 | 71.3 | 75.5 |
| TPH-YOLOv5 [16] | 80.8 | 82.2 | 66.8 | 79.5 | 71.6 | 75.2 |
| Proposed | **84.3** | 82.8 | **67.0** | **84.6** | 72.0 | **77.3** |

### 4.4.2. Results from the VEDAI Dataset

The VEDAI dataset contains a variety of small objects in remote sensing images with different light and angular transformations. In our experiments, we compare the proposed HRTP-Net algorithm with other state-of-the-art algorithms, and the results are shown in Table 5. Our algorithm reaches the highest in the classes of plane, pick_up, tractor, truck and in the mAP of all classes compared to YOLOv5 as the baseline with a 6.8% improvement. The pick_up and tractor class objects are very similar to the truck category objects, but HRTP-Net is also able to identify these two classes effectively with mAPs of 76.1% and 58.6%, respectively. Figure 9 shows some of the detection results of our algorithm in the VEDAI dataset.

**Table 5.** Comparison of detection results from the VEDAI dataset (The best performance is highlighted with bold font).

| Model | Plane | Boat | Camping_car | Car | Pick_up | Tractor | Truck | Van | Others | All Classes |
|---|---|---|---|---|---|---|---|---|---|---|
| | mAP (%) | | | | | | | | | mAP (%) |
| YOLOR [33] | 45.8 | 10.5 | 54.7 | 74.0 | 64.4 | 30.3 | 17.4 | 11.1 | 13.1 | 35.7 |
| YOLOv4 [34] | 67.0 | 38.7 | 61.0 | 85.5 | 74.0 | 48.6 | 60.4 | 15.0 | 28.7 | 53.2 |
| YOLOv5 [37] | 78.4 | 25.3 | 68.3 | 82.6 | 74.1 | 40.4 | 53.3 | 27.3 | 30.5 | 53.4 |
| YOLOv7 [35] | 25.6 | 11.4 | 56.2 | **87.9** | 75.9 | 56.0 | 30.5 | 10.5 | 11.5 | 36.5 |
| PicoDet [36] | 44.6 | 19.1 | 61.7 | 79.1 | 61.0 | 31.1 | 32.7 | 13.5 | 18.2 | 40.1 |
| Faster R-CNN [5] | 65.9 | 53.2 | 68.0 | 60.6 | 61.8 | 45.5 | 55.8 | 17.3 | **51.7** | 53.3 |
| Cascade R-CNN [38] | 53.2 | **68.4** | 70.3 | 72.0 | 67.5 | 40.8 | 49.0 | **39.7** | 50.8 | 56.9 |
| HRNet [11] | 42.3 | 62.9 | 65.5 | 77.4 | 74.0 | 44.0 | 45.9 | 17.9 | 42.4 | 52.5 |
| Super YOLO [44] | 58.7 | 28.5 | **73.3** | 87.5 | **80.1** | 52.3 | 51.7 | 37.8 | 21.8 | 52.5 |
| PAG-YOLO [43] | 74.7 | 37.7 | 68.4 | 82.5 | 72.8 | 43.3 | 53.8 | 14.0 | 31.8 | 53.2 |
| TPH-YOLOv5 [16] | 71.8 | 31.1 | 62.1 | 81.9 | 70.6 | 41.4 | 55.4 | 40.2 | 35.6 | 54.5 |
| Proposed | **88.2** | 47.5 | 61.6 | 83.9 | 76.1 | **58.6** | **62.9** | 36.8 | 26.5 | **60.2** |

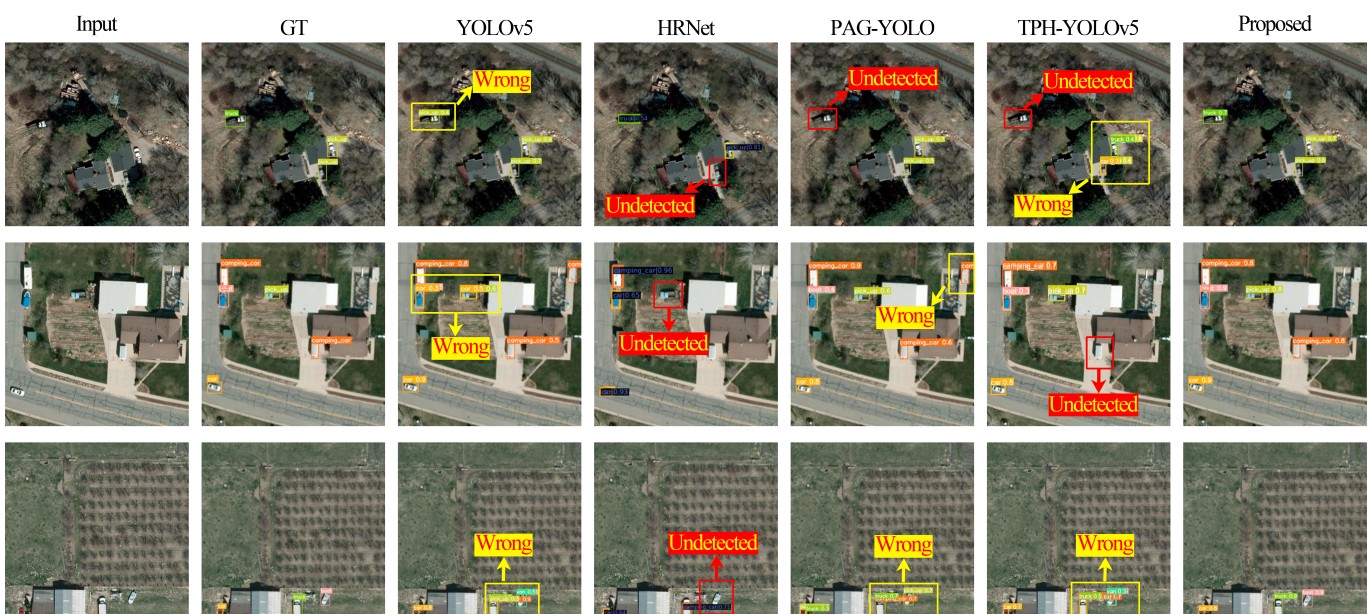

**Figure 9.** Selected examples from the detection results from the VEDAI dataset obtained by HRTP-Net.

### 4.4.3. Results from the DOTA Dataset

In the DOTA dataset, there are many very small and dense objects. Compared with other algorithms, HRTP-Net is able to outperform the other algorithms and achieve an mAP value of 73.7%, shown in Table 6. Our algorithm achieves the highest mAP in the small-vehicle class with 61.6% because this class contains more small objects. Figure 10 shows the visualization of the proposed algorithm in the DOTA dataset, and we find

that the HRTP-Net still encounters issues with missed and false detections when faced with dense objects. This may be due to the fact that the window operation in the W-trans structure separates the dense objects. Meanwhile, for rotating objects, the horizontal bound boxes can make it difficult for the model to learn accurate features. These are the problems we need to focus on in our future work.

**Table 6.** Comparison of detection results inn the DOTA dataset (The best performance is highlighted with bold font).

| Model | Small Vehicle | Ship | All Classes | | |
| --- | --- | --- | --- | --- | --- |
| | mAP (%) | | Precision (%) | Recall (%) | mAP (%) |
| YOLOR [33] | 60.4 | 84.3 | 85.3 | 68.5 | 72.3 |
| YOLOv4 [34] | 59.3 | **86.7** | 84.4 | 67.0 | 73.0 |
| YOLOv5 [37] | 58.5 | 86.2 | 84.0 | 68.5 | 72.4 |
| YOLOv7 [35] | 59.7 | 86.5 | 86.1 | 67.1 | 73.1 |
| PicoDet [36] | 59.9 | 84.9 | 85.9 | 65.9 | 72.4 |
| Faster R-CNN [5] | 26.0 | 53.3 | 64.6 | 41.8 | 39.6 |
| Cascade R-CNN [38] | 42.0 | 53.4 | 62.2 | 48.7 | 44.7 |
| HRNet [11] | 26.4 | 53.9 | 71.9 | 43.4 | 40.1 |
| SuperYOLO [44] | **62.9** | 83.5 | 85.9 | 68.9 | 73.2 |
| PAG-YOLO [43] | 58.4 | 86.5 | 84.3 | 68.8 | 72.4 |
| TPH-YOLOv5 [16] | 59.8 | 86.2 | **87.4** | 65.0 | 73.0 |
| Proposed | 61.6 | 85.7 | 84.5 | **69.8** | **73.7** |

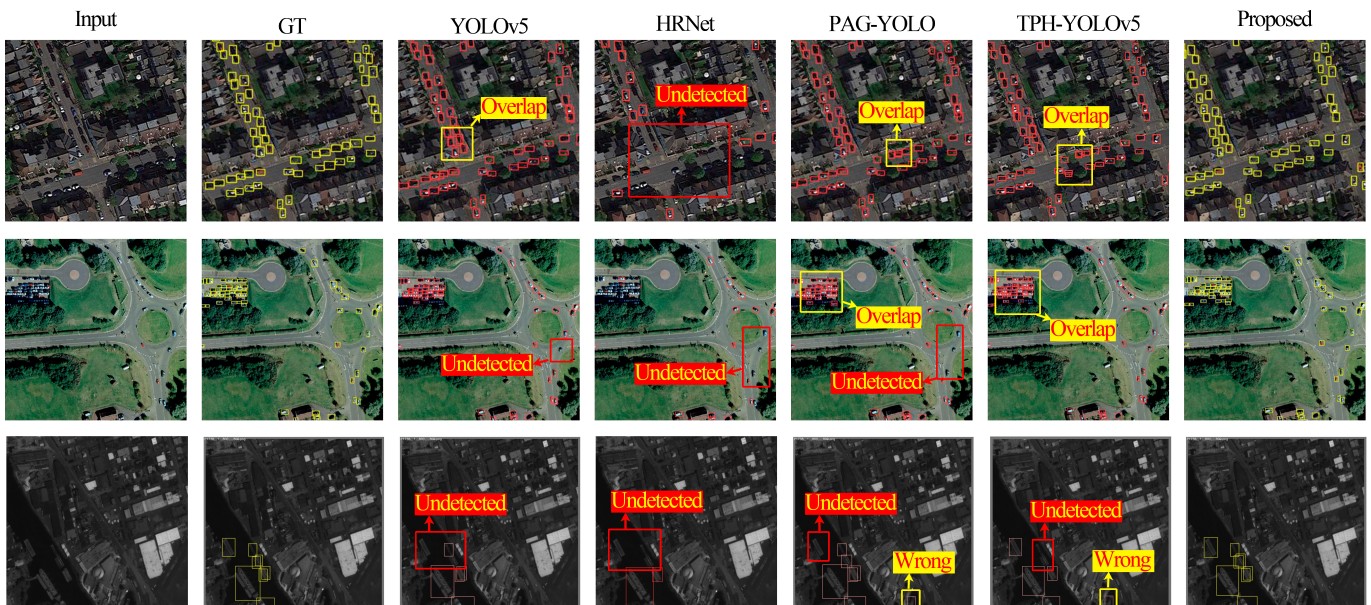

**Figure 10.** Selected examples from the detection results in the DOTA dataset obtained by HRTP-Net.

### 4.4.4. Efficiency Analysis

For a comprehensive comparison, Table 7 shows the speeds and parameters for all models in the same operating environment. Frames per second (FPS) in the table indicates the number of images processed by the model per second. Generally speaking, two-stage models such as Faster R-CNN, Cascade R-CNN, and HRNet are lower on FPS than one-stage models, and our proposed HRTP-Net is faster than the three two-stage models. In terms of the number of parameters, the smallest one is PicoDet with only 1.35 M. In terms of time complexity, our model contains more convolution operations and, therefore, has a higher GFLOPs than the existing one-stage models but still has less computational time compared to the two-stage models. The fastest detection speed is YOLOv5 with 256 FPS in the MASATI dataset. Our model is larger than the fastest YOLOv5 model in terms of the

number of parameters, which may limit the application of our proposed model in certain scenarios as small mobile devices.

**Table 7.** Performance comparison of different methods in the MASATI dataset.

| Methods | GFLOPs | Params (M) | FPS |
|---|---|---|---|
| YOLOR [33] | 119.3 | 52.49 | 84 |
| YOLOv4 [34] | 20.9 | 9.11 | 152 |
| YOLOv5 [37] | 16.3 | 7.05 | 256 |
| YOLOv7 [35] | 103.2 | 36.48 | 109 |
| PicoDet [36] | 13.7 | 1.35 | 175 |
| Faster R-CNN [5] | 91.0 | 41.12 | 109 |
| Cascade R-CNN [38] | 118.81 | 68.93 | 73 |
| HRNet [11] | 83.19 | 27.08 | 56 |
| SuperYOLO [44] | 31.59 | 7.051 | 84 |
| PAG-YOLO [43] | 16.4 | 7.10 | 222 |
| TPH-YOLOv5 [16] | 15.5 | 7.07 | 192 |
| Proposed | 81.9 | 16.58 | 119 |

## 5. Discussion

Through ablation experiments and comparisons with state-of-the-art models, our model's advancements in the task of small object detection in remote sensing images are demonstrated. We present the detection performance of HRTP-Net in various challenging scenarios. In Figure 11a, the object occupies only a few pixels with insignificant visual features. Our model utilizes HR-FFN to preserve detailed information during forward propagation, resulting in excellent recognition performance for small objects. In Figure 11b, the complex water around the object significantly affects feature extraction. However, our proposed STMA structure effectively highlights the object and suppresses the influence of complex backgrounds, leading to accurate detection. In Figure 11c, the objects are covered by different illumination conditions and backgrounds, degrading the visual features. Nevertheless, our algorithm effectively handles the detection task in complex backgrounds, enabling precise recognition of the objects. In Figure 11d, the large number of objects and the presence of similar objects pose a significant challenge to detection precision. However, our proposed algorithm performs well in handling complex backgrounds and accurately identifying small-scale car targets in urban scenes.

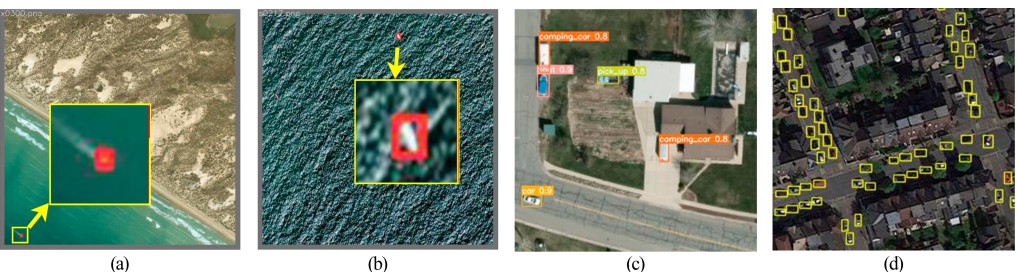

(a)      (b)      (c)      (d)

**Figure 11.** Test results of the algorithm under different challenges. (**a**) An extremely small object is in the bottom left corner of the image. (**b**) The ship object is surrounded by complex water waves. (**c**) Objects are disturbed by illumination variation and shadows. (**d**) There is similar interference around objects.

However, our algorithm still has certain limitations. First, our algorithm currently can only detect objects with horizontal bounding boxes. This limitation leads to decreased recognition accuracy when dealing with a large number of dense objects or rotated bounding box objects due to overlapping detection boxes. Second, when facing multi-scale objects, particularly large-scale objects, the proposed STMA in our algorithm employs the window-based mechanism to segment feature maps, resulting in incomplete object features and

ultimately causing failures in object detection. These two shortcomings will be the direction of our future research, as we aim to delve deeper into addressing these limitations.

In addition, we have also applied the research findings of this paper to other algorithms in different domains. For the task of multi-object tracking in remote sensing videos, the algorithm achieves object tracking through a two-stage process: first, detecting the objects in each frame and, then, tracking them based on the detection results. We propose an algorithm called "HRMOT: Two-step association-based Multi-object Tracking in Satellite Videos Enhanced by High-Resolution Feature Fusion", which was published in BICS 2023. This algorithm combines YOLOv5 with the high-resolution network (HRNet) to enhance the tracking performance by improving the accuracy of small object detection, leveraging the research findings from our HRTP-Net model. Therefore, we believe that the research on HRTP-Net in this paper can also be applied to other tasks in the remote sensing field.

## 6. Conclusions

In this paper, we aim to obtain accurate details in remote sensing images to improve the feature effectiveness of small objects. We propose an approach called HRTP-Net with a high-resolution network and a parallel detection structure by combining self-attention and spatial attention mechanisms. Specifically, the proposed HR-FFN is capable of generating a more accurate object location with enhanced features by maintaining multi-scale high-resolution features with rich semantic information. Moreover, in STMA, the Swin transformer is utilized to calculate the global information and attached with a pixel-level attention mechanism to achieve background–object modeling. In addition, the parallel detection structure leverages the attentional outputs of STMA to guide the main detection structure to obtain discriminative features. We conducted experiments on three extensively used datasets. The experimental results demonstrate that our method outperforms widely used methods with a real-time detection in the MASATI dataset, the VEDAI dataset, and the DOTA dataset.

**Author Contributions:** Conceptualization, X.Z.; methodology, X.Z. and Q.L.; software, X.Z.; validation, H.C. and Q.L.; formal analysis, X.Z.; investigation, X.Z. and Q.L.; resources, H.S.; data curation, X.Z. and H.C.; writing—original draft preparation, X.Z.; writing—review and editing, Q.L. and H.S.; visualization, X.Z.; supervision, H.S.; project administration, H.S. and Q.L.; funding acquisition, H.S. All authors have read and agreed to the published version of the manuscript.

**Funding:** This work was supported in part by the 2023 Jilin Province and Chinese Academy of Sciences cooperative high-tech industrialization project for the specialized program under Grant 2023SYHZ0010.

**Acknowledgments:** This work was supported in part by the Jilin Province Key Laboratory of Machine Vision and Intelligent Equipment.

**Conflicts of Interest:** The authors declare no conflict of interest.

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
