# Peer review of "High-Resolution Network with Transformer Embedding Parallel Detection for Small Object Detection in Optical Remote Sensing Images"

_remotesensing, doi:10.3390/rs15184497_

Round 1
Reviewer 1 Report
The manuscript discusses a novel parallel detection structure and the utilization of self-attention in HRTP-Net (good work). Could you perhaps provide an explanation of how this particular mix of methods and techniques contributes to the progress and improvement of small item detection in the field of remote sensing? What are the potential practical ramifications and real-world applications that can be foreseen as a result of HRTP-Net's enhanced performance?
Comment Attached#
Best regards,

The overall quality of English language in the manuscript is acceptable, but there are Some sentences that are not as clear. I recommend the authors carefully review and revise those sentences to enhance the overall clarity and coherence of the paper.
Reviewer 2 Report
The study is designed and well-explained in a way that will interest readers. However, the following corrections would make the study more understandable.
Line 42-43: Figure 1 is not precise. Please increase the DPI.
Line 173-174: Figure 2 is not precise. Please increase the DPI.
Line 311: The 512 × 512 images???? I think you mean pixels.
Reviewer 3 Report
The problems with the paper are as follows:
1." First, the down- sampling operation commonly used for feature extraction can barely preserve weak features of objects in a tiny size. Second, convolutional neural network methods have limitations in modelling global context to address clutter backgrounds."How does the method proposed in this article address these two limitations? The following content did not answer this question. It is recommended that the author pay attention to the coherence of the language in the paper.
2.How to define small objects in remote sensing images? Suggest the author to provide a clear definition in the main text.
3.Figure 2 should be under the title of 3.1, not above. Meanwhile, it is recommended to mark (1) , (2),(3),(4) in Figure 2 for the author's understanding.
4.The text in Figure 9 is not clear, it is recommended that the author modify it.
5.There are grammar errors in the paper. It is recommended that the author polish the entire text,For example,“A detection approach for small object ”should be ”A detection approach for small objects”
Minor editing of English language required.
For example,“A detection approach for small object ”should be ”A detection approach for small objects”
Round 2
Reviewer 1 Report
Dear authors, thank you for carefully consideration the comments in your manuscript: High-Resolution Network with Transformer Embedding Parallel Detection for Small Object Detection in Optical Remote Sensing Images., which improved the quality of your work presentation. (good job);
Best regards,
Reviewer 3 Report
accept
Minor editing of English language required